# Technological Advancements and Elucidation Gadgets for Healthcare Applications: An Exhaustive Methodological Review-Part-II (Robotics, Drones, 3D-Printing, Internet of Things, Virtual/Augmented and Mixed Reality)

**Sridhar Siripurapu [1], Naresh K. Darimireddy [1], Abdellah Chehri [2,\*], Sridhar B. [1] and Paramkusam A.V. [1]**

1   Lendi Institute of Engineering & Technology, Jonnada 535005, India
2   Department of Mathematics and Computer Science, Royal Military College of Canada, Kingston, ON 11 K7K 7B4, Canada
\*   Correspondence: chehri@rmc.ca

**Abstract:** The substantial applicability of technological advancements to the healthcare sector and its allied segments are on the verge of questioning the abilities of hospitals, medical institutions, doctors and clinical pathologists in delivering world class healthcare facilities to the global patient community. Investigative works pertinent to the role played of technological advancements in the healthcare sector motivated this work to be undertaken. Part-I of the review addressed the applicable role play of advanced technologies such as Artificial intelligence, Big-data, Block chain, Open-Source and Cloud Computing Technologies, etc., to the healthcare sector and its allied segments. The current Part-II manuscript is critically focused upon reviewing the sustainable role of additional disrupting technologies such as Robotics, Drones, 3D-Printing, IoT, Virtual/Augmented/Mixed Reality, etc., to uncover the vast number of implicit problems encountered by the clinical community. Investigations governing the deployment of these technologies in various allied healthcare segments are highlighted in this manuscript. Subsequently, the unspoken challenges and remedial future directions are discussed thereof.

**Keywords:** robotics; drones; 3D-printing; internet of things; virtual/augmented and mixed reality

## 1. Introduction

The past few decades witnessed extensive research in the field of healthcare. Services such that technological advancements are indispensable in the healthcare domain and its allied segments—apart from the various pulmonary (viral), bacterial and inflammatory diseases (including the deadly COVID-19 pandemic) challenging the medical world and critically exposing the limitations of existing health practices [1–3]. Plenty of viruses found in air, water, soil, etc., are causing different infectious diseases, from the flu and cold to the deadly COVID-19 pandemic, ultimately leading to multi-organs failure followed by cardiac arrests [4–7]. The sudden outburst of the COVID-19 pandemic also uncovered the deficiencies of global health-care systems in handling public-health emergency scenarios coupled with plenty of uncovered ailments in clinical microbiology, tuberculosis diagnosis, preoperative surgical planning, Spine Medicine, medical education, mastectomy surgical planning, Orthopaedic Surgery, Laparoscopic Surgery, Surgical Training, Robotic Surgeries, Anxiety and Depression treatments and Hospital Navigations, etc., motivating the need for identifying the deployment of technological advancements addressing them. Therefore, this review work is conducted in two parts, typically highlighting the contribution of technologies such as AI, Big-data, Block-chain Technology, Open-Source Technologies and Cloud computing, etc., in Part-I—while technologies such as Robotics, Drones, 3D-Printing, IoT (Internet of things), Augmented/Mixed Reality, etc., and their

role played in uncovering many healthcare issues, are discussed in the current Part-II of the manuscript. Purposeful technological implications, such as wireless communication gadgets, smart intelligent devices, etc., are deeply embedded in every aspect of treating patients, right from the simplest diagnostic scenario to the complex life threatening cases, through the instant disbursement of patient data (Bigdata) and diagnostic information to physicians around the world; robotic technology is deployed into tele-health facilities, educational institutions, super specialty hospitals, surgical ambulances, diagnostic and medical laboratories, etc. [1]. Robots are considered viable technological solutions for healthcare problems [2], typically revolutionizing the ways and means of delivering patient care by performing mundane or dangerous tasks with deeply penetrated grassroots in the health care sector. In the case of natural hazards and pandemics, robotic systems are also working in collaboration for the rescue of medical teams and doctors. Varieties of robotic systems are investigated to support clinicians and radiologists in the treatment of heart stroke and brain injuries, etc. Further robotics tackling emotional responses (to ensure the long-term care of patients across the world), surgical performances, remote health monitoring activities, medication handling, pharmaceuticals, Tele-health, physical therapies, surgical ambulances and medical diagnostic laboratories are a few significantly allied areas of healthcare propounded by robotics.

Remotely controlled multipurpose autonomous aerial vehicles (drones) are being optimally utilized to carry the healthcare payload (typically in this case), delivering feasible solutions to an innumerable number of healthcare problems by reducing the transport overheads and infrastructural deficiencies, thereby enhancing the critical life-giving health-care services, including the facilitation of supply chain logistics [3] to individuals located in remote areas. Drones are also deployed for telemedicine applications, ensuring the diagnosis and treatment for patients located remotely.

Meanwhile, the Additive manufacturing (or 3D-printing) technology is envisioned to print medical and lab equipment with the third dimension, thereby equipping the medical community with the flexibility to critically analyses complex medical scenarios akin to the healthcare sector. Three dimensional printing ensures the printing (development) of novel surgical cutting (and drill) guides, customized prosthetics and implants tailored to individuals as well as creates patient specific replicas of bones, organs and blood vessels within no time. Thereby, this provides a wide scope for medical professionals to understand the inner anatomy of patients and enhance their preparation levels to accept the treatments. Today, 3D-printing is taken advantage of by the cardiac, orthopaedic, vascular, neurosurgical physicians and surgeons to accurately plan the surgical procedures coupled with cross-sectional imaging (or modelling of surgical tools) based on patient-specific anatomy. As far as regenerative medicine and tissue engineering is considered, 3D bio-printing is used to create living human cells and (or) tissues.

On the other hand, Internet of Things (IoT) technology is proven to be one of the most reliable technological standards, typically operating with sensors such as pulse rate sensor, temperature sensor, accelerometer, and respiration sensor, etc., and is utilized in various domain sectors, from regular usage products to the industrial monitoring systems and patient wearable healthcare gadget-based smart health monitoring systems [4]. Such monitoring systems typically help the clinical community to continuously monitor the health conditions of patients suffering from life threatening diseases—by posting them, they are updating real time health information to doctors [5]. The widespread access of mobile internet provision coupled with healthcare service systems supported by the Android open source design [6] and Raspberry Pi suffice the right IoT platforms for healthcare monitoring more succinctly—plenty of sensors such as the wireless sensor, Bluetooth module, Fingerprint sensor, Gyroscope, Magnetometer, Barometer, Proximity, GPS tracker, Camera, NFC-near field sensor, etc., positioned in the smart-phones of today are widely used in the development of healthcare monitoring systems [7].

Further, the applicability of the virtual environment-based Immersive Technologies such as VR(Virtual Reality), AR(Augmented Reality) and MR(Mixed Reality) carried with

them unlimited possibilities in healthcare applications, delivering realistic physical experiences through immersive real life simulations pertinent to surgical procedures, medical therapies, preventive medical education and training, typically providing clinical pathologists and doctors with a visualized view of the patient's internal anatomy without involving any invasive procedures. Archaic numbers of the investigations are under progress to innovate an uncountable number of privacy-sensitive technological gadgets in order to assist the clinical specialists and medical surgeons globally to set right many health disorders and ailments, thereby helping government organizations in safeguarding the lives of people through the deployment of timely medication measures.

The rest of the manuscript is organized as follows: Section 2 provides a methodical review of emerging technologies such as Robotics, Drones, Three-Dimensional Printing, Internet of Things and Virtual Reality/Augmented Reality/Mixed Reality (HoloLens), etc., and their relevant framework solutions analogous to a vast number of healthcare segments. Meanwhile, Section 3 highlights the implicit pros and cons associated with the applicability of these technologies to the healthcare issues, typically highlighting the unspoken challenges and leaving room for the interested research community to explore them. Section 4 presents the concluding remarks.

## 2. Disrupting Technologies

### 2.1. Robotics

Today, autonomous robotic systems are employed in almost every sector, providing long-term care solutions [8] to many contemporary issues from research centres to the local hospitals, including operation theatres to clinical settings, rendering exemplary support to various healthcare segments of the clinical arena. Advancements in electronics and communication engineering, on the other hand, are directly attributed to wide research publications, depicting a marginal shift in robotics from industrial platforms to the day-to-day healthcare contexts of the global community. In the medical sector, robots are helping to improve the operational efficiencies by taking over low value administrative (and/or repetitive) clinical tasks and updating Electronic Health Records (EHR) of patients—converting medical procedures into more feasible and economically viable ones rather than being inefficient. The evolution of humanoid robots into the healthcare sector signified the positive impact of robotics' deployment in the healthcare sector, aiding clinicians and pathologists. Today, robotics developed a strong integration with Virtual Reality [9] and further heading towards integration with Artificial Intelligence (AI) [10], research works demonstrating the deployment of various healthcare robots (Figure 1) include:

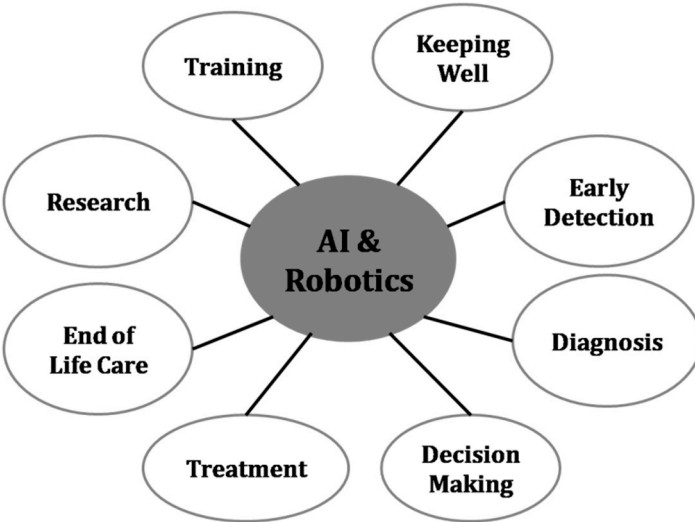

**Figure 1.** AI and Robotics for healthcare.

(A) Surgical Robots—Surgical robots offer high-definition 3-D view capabilities promoting the deployment of robots in "minimally invasive" Complicated Robotic surgeries—Da Vinci Surgical System is one such robotic laparoscopic surgical platform in the United States since 2000. Today, many Global Big Wig companies such as Stryker, Globus Medical, Johnson & Johnson, Siemens Stereotaxis, Smith & Nephew, Mazor Robotics, Auris Health, Intuitive Surgical, Zimmer Biomet, and Medtronic, etc., have developed surgical robots [11] and highlighted the utility of robotics in surgical systems, laparoscopy surgery and tele-rounding robots [12], robotic rehabilitation [13] and dentistry applications, etc.

(B) Clinical training robots—to enhance the knowledge of healthcare providers by providing realistic simulation-based training. For example, Paediatric Hal is an AI-based training robot manufactured by Gaumard Scientific [14,15].

(C) Pharmacy automation robots—for dispensing pharmaceuticals (medicines) at high speed in any medical environment.

(D) Disinfection robots—Pulsed ultraviolet light-based robots disinfect the entire room in a few minutes to prevent the spread of infections [16].

(E) Companion robots—'Care bots' to engage with patients emotionally.

(F) Caring robots—Humanoid Robots for transferring aged people from the bed to the wheelchair and back [17].

(G) Tele-presence robots—Tele-operated robots allow healthcare people to work safely from infectious patients: For example, Da-Vinci Robot [18].

(H) Wheeled robots—Mobile robots to move in an environment instead of remaining fixed [19].

(I) Flying mobile robots—Flies like Aerial drones—Quad copters category.

(J) Legged robots—Contain articulated legs to provide locomotion on the ground.

(K) Wearable robots—Wearables to measure body signals and augment human capabilities.

Exoskeleton Robots, Ambulance Robots, Receptionist Robots, Physical Therapy Robots, Rehabilitation Robots and Nursing Robots, etc., are other segments of healthcare applications that witnessed the penetration of robotics (Figure 2).

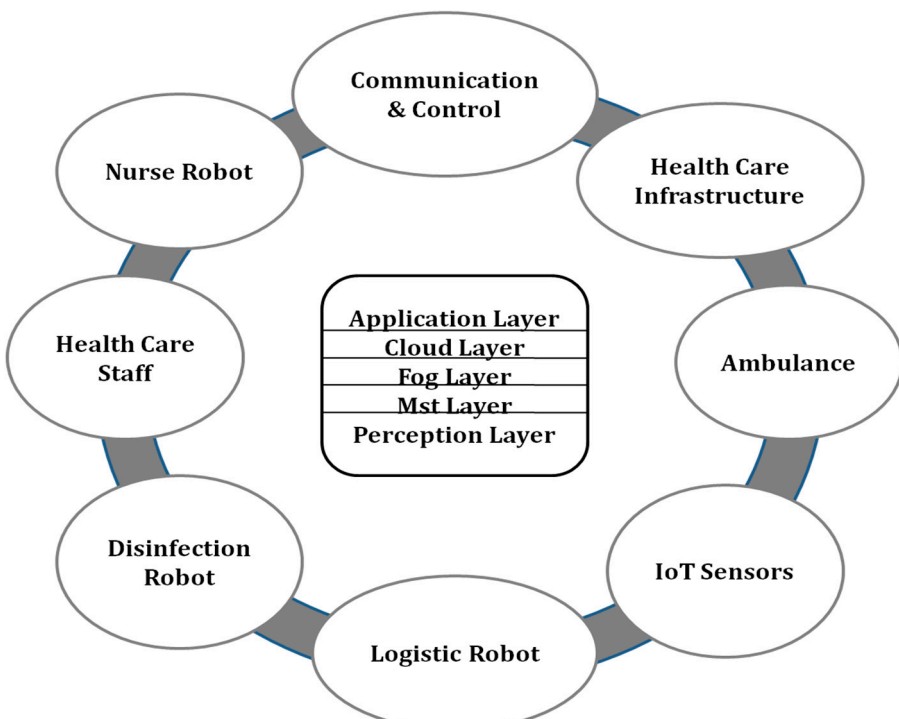

**Figure 2.** Robotics for healthcare applications.

Robots with AI-enabled medicine identifier softwares drastically reduce the time taken to distribute medicines as well as to perform accurate surgeries in tiny places. Explorative investigations depicting the applicability of robotics are tabulated below in Table 1.

**Table 1.** Review of applicability of robotics.

| Sl. No | Author | Ref. | Category of Health Segment | Application Type | Year |
|--------|--------|------|----------------------------|------------------|------|
| 1 | Giovanni et al. | [19] | | Survey on rehabilitation robotics | 2022 |
| 2 | Tietze et al. | [20] | | Barriers of assistive robotics | 2021 |
| 3 | Al-Rawabdeh et al. | [21] | | Robotics targeting neural rehabilitation | 2021 |
| 4 | Oña et al. | [22] | Robotics | Case study of robotics in nursing | 2020 |
| 5 | Tzafestas et al. | [23] | | Robot interventions in rehabilitation of upper limbs | 2019 |
| 6 | Agnihotri et al. | [24] | | Ethics involved in applicability of robotics and automation | 2018 |
| 7 | Chehri et al. | [25] | | Robotics in nursing | 2018 |

### *2.2. Drones*

Health-care providers started deploying AI-based robotic health care systems in performing virtual diagnosis and suggesting predictive medications in a few urban segments of the society; on the other hand, the Unmanned Autonomous Remotely Operated Aerial Vehicles (UAVs, i.e., Drones) are employed to solve the major constraints such as availability of drugs, vaccines, blood samples and even collection of biological samples in many under developed countries [26–28]. Drones "Add Eyes", i.e., help technology to explore the uncovered segments of people deprived of healthcare facilities by providing timely healthcare services to individuals at unreachable locations (Figure 3), thereby promising lifesaving medication facilities to the people [29].

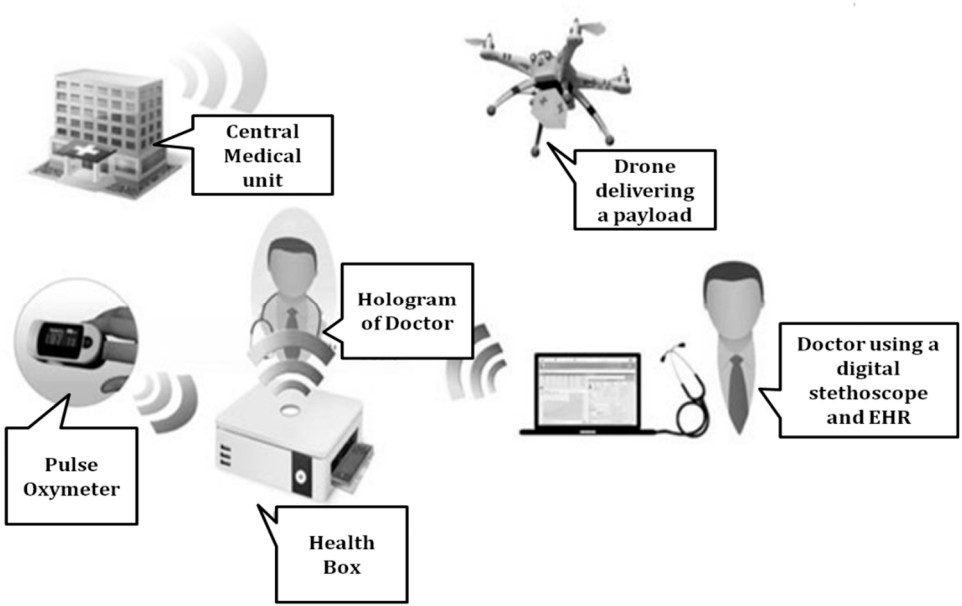

**Figure 3.** Deployment of drones in healthcare applications.

In the present-day scenario, UAVs totally erased the distance barriers in the healthcare sector by reducing the mortality rate cause by diseases such as dengue, postpartum hemorrhage conditions, blood loss in accident cases and cases demanding timely critical organ grafting, etc.; by adopting sky-based advanced logistics systems, drones could deliver

blood samples and vaccines to the healthcare institutions and sometimes directly to the residence of patients [30–33]. The majority of the developed countries deployed drones in public health departments to ensure better patient caring [34–36]. Telecommunication drones are used for preoperative evaluation and tele-mentoring in remote areas [37]. A typical utilization of drones for various healthcare applications in a few countries is tabulated (Table 2) below.

**Table 2.** Analysis of drones' utility in healthcare applications.

| Drone Company | Health Items | Delivery Location |
|---|---|---|
| Matternet | Blood, medications | Haiti, Domnican Republic, Papua New Guinea, Switzerland |
| DHL Parcel | Blood, medications | Germany |
| Zipline | Vaccines, blood | Rwanda |
| Flirtey | Medications | Virginia, Nevada |
| Delft University | Defibrillators | Netherlands |

In spite of the worldwide acceptability of drones in the healthcare sector, India being a massive country with equi-differentiable geographical and health-care disparities issued UAV guidelines and a framework for a Drone Ecosystem Policy Roadmap to implement drone delivery solutions for medical and healthcare applications such as Pre-hospital Care, Laboratory Testing, vaccine and blood delivery [21,38–40], etc. Environmental monitoring (e.g., wildfire, landslide, and air quality monitoring) with drones [41–47], drones to deliver automated external defibrillators (AEDs) for cardiac emergencies [48–53], drones for transporting biological samples [54–57], drones for search and rescue operations [58,59] and emergency service delivery [60,61], are a few other subjectively applicable areas of drones. Earmarking the roundabout utilization of drones, the government of India in association with other states formulated a pilot project titled 'Medicines from the Sky' to investigate the increasing utility of drones in the healthcare sector [62]. The global market for drone's package delivery is anticipated to reach USD 7380 million by 2027, with around 14 key companies profiled from the US, South Africa, Israel, Canada, and Germany [63]. Though plenty of explorations by eminent researchers pertaining to the applicability of UAVs in numerous healthcare segments are under progress, few of them are tabulated (Table 3) in addition to those already discussed above.

**Table 3.** Review signifying utility of drones in healthcare domain.

| Sl. No | Author | Ref. | Category of Health Segment | Application Type | Year |
|---|---|---|---|---|---|
| 1 | Gupta et al. | [64] | | Medical drones in healthcare delivery | 2021 |
| 2 | Mehta et al. | [65] | | Block chain and drone-based health delivery scheme | 2021 |
| 3 | Ahmed et al. | [66] | | Role of AI-drones in Indian cities' digitalization | 2021 |
| 4 | Angurala et al. | [67] | | Drones in clinical microbiology and infectious diseases | 2020 |
| 5 | Sedig et al. | [68] | | IoT-based drones to prevent spread of COVID-19 | 2020 |
| 6 | Uttam et al. | [69] | Drones | Study on public perception of drones to deliver external defibrillators | 2020 |
| 7 | Hiebert et al. | [28] | | Drones in healthcare services | 2020 |
| 8 | Cawthorne et al. | [70] | | Drones for tuberculosis diagnosis | 2020 |
| 9 | Mc Call et al. | [71] | | Framework for drones' usage in healthcare | 2020 |
| 10 | Robakowska et al. | [72] | | Africa: Medical drones | 2019 |
| 11 | Guillen-Perez et al. | [73] | | Medical applications of drones | 2018 |
| 12 | Chehri et al. | [74] | | Flying ad hoc networks | 2018 |

### 2.3. Three Dimensional Printing

"Organovo and Envision TEC—Pioneers of 3D Printing Technology".

Three dimensional printing typically creates 3D (three dimensional) objects by stacking layers of raw materials such as metals, plastics, and ceramics rendered from digital files of magnetic resonance images (MRI) and computer-aided design (CAD) drawings, etc. The 3D-printing process experienced a phenomenal expansion in recent years since its first commercialization in year 1980 by Charles Hull [2]. Presently, 3D-printing is offering noteworthy benefits to patients and clinicians in manufacturing precise and personalized pharmaceuticals to produce highly customized products in healthcare applications (Figure 4).

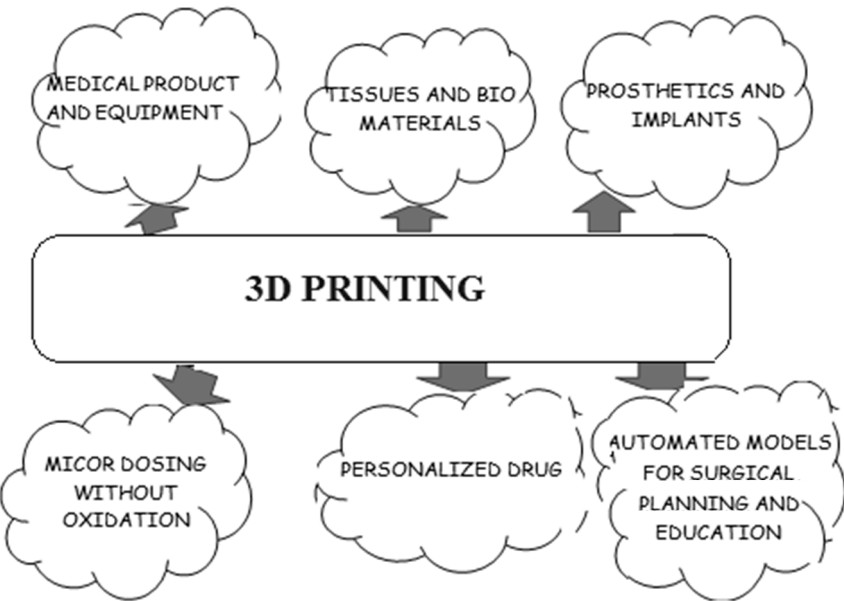

**Figure 4.** 3D-printing and its applications.

Three dimensional printing is also deployed in craniofacial, plastic, urology, and dental surgery fields, whereas its utility in orthopaedics is limited because of the lengthy operation times, intra-operative bleeding, prolonged time of anaesthesia and large doses of medications, etc. [75]. There are plenty of experimental investigations and evaluations pertaining to the utility of 3D-printing for healthcare applications under progress; a few of them include: applications of 3D-printing for hearing problems, dentistry jaw replacements, provision of bones, crowns, and bridges, limb replacement, and tooth caps [76]; 3D-printing utility in medical and dental fabrication; customized implants [77], prosthetics [78], medical models [79], medical devices [80], and dentistry [81]; a summary of 3D-printing applications in medicine can be found on the YouTube channel [82]. Chuck Hull [83,84], the founder of the 3D Systems Corporation, developed models for generating 3D objects [85–87], 3D modelling of foetal heart defects [88], and glasses for the visually impaired [89]. Envision TEC and Denmark-based Widex [90] developed 3D-processed Individual Shells for Hearing Aids and Poland Institute of Pathology used 3D modelling for teaching purposes [91]. Three dimensional printers were used for dental applications [92–95], 3D-printed implants for treatment of diabetes [96], arthritis [97], implants for crippled animals [98], etc. The first 3D-printed drug is called Sprintam (levetiracetam, for epilepsy treatment) [99].

The advancements of 3D-printing in medical technology are boundless, such that today a radiologist can comfortably imitate a patient's spine in no time, while a dentist can customize crown (dental) restorations matching the patient's anatomy; the production of 3D-customized prosthetic limbs, cranial implants and/or orthopaedic implants akin to knees and toes are brought into reality. Today, healthcare professionals, research organizations, and hospitals across the globe are using patient-specific 3D-printed tactile anatomical

reference models (derived from patients' scan data) for preoperative planning during operational procedure and postoperative planning procedures, etc. Above all—other healthcare 3D printable subjective areas include:

(A) Production of new medical products → Manufacturers of medical tools and product companies adopted 3D-printing technology to accurately fabricate prototypes of brand new medical devices and surgical instruments.

(B) Tissue Engineering and 3D-printed organs (bio-printing) → Today, researchers of bio-printing are recreating precisely shaped and geometrically sized synthetic blood vessels (and organs on demand)—totally eliminating the need for autografts in the future. One typical scenario depicted a 3D-printed anatomical model of a hand depicted (Figure 5) with skin grafted as well.

(C) Economically feasible prostheses → 3D-printing technology is playing an optimally promotable role to produce prosthesis sockets for hundreds and thousands of people deprived of access to prosthesis treatments due to financial barriers.

(D) Timely insoles/orthoses → 3D-printing technology erased delays associated in patient-specific insoles and orthoses to ensure improved physical therapies, etc.

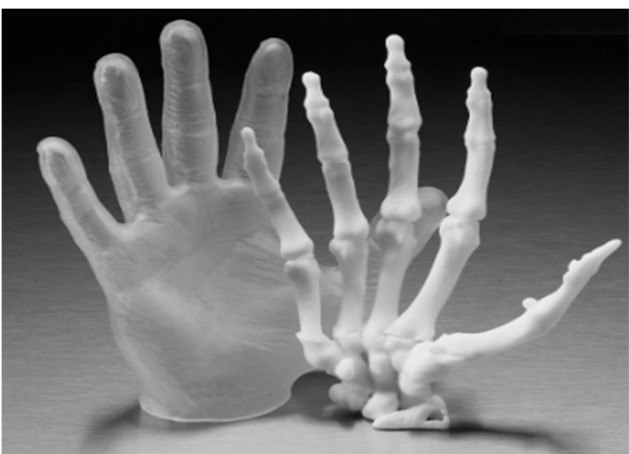

**Figure 5.** 3D anatomical model of hand–skin made of elastic 3D-printing material.

Many other investigations committed to enlighten the global world about the applicability of 3D-printing in various healthcare segments are shown in Table 4.

**Table 4.** Review signifying applicability of 3D-printing in various healthcare segments.

| Sl. No | Author | Ref | Category of Health Segment | Application Type | Year |
|---|---|---|---|---|---|
| 1 | Longhitano et al. | [100] | | Review on role of 3D-printing during COVID-19 | 2021 |
| 2 | Aimar et al. | [101] | | 3D-printing in medical applications | 2019 |
| 3 | Mardis et al. | [102] | | | 2018 |
| 4 | Shahrubudina et al. | [103] | 3D-printing | Overview of 3D-printing technology, materials, and applications | 2019 |
| 5 | Vaish et al. | [104] | | Applications of 3D-printing in orthopaedics | 2018 |
| 6 | Dodziuk et al. | [105] | | Applications of 3D-printing in healthcare | 2016 |

### 2.4. IoT

The future of medical 3D-printing technology was presumed to diversify beyond scaffolds over the past ten to fifteen years to the extent of people walking around with 3D-printed organs, i.e., using their own printed cells, rather than those transplanted from others. Numerous 3D-printing tools and therapeutic methods developed today add

new degrees of comfort and personalized treatment to patients, thereby equipping the clinical community (doctors) with greater insights for understanding complex cases and planning for meticulous treatments. On the contrary, the Internet of things (IoT) technology demonstrated a prominently viable role in connecting various medical devices, sensors, and healthcare professionals to deliver superior quality medical services (Figure 6) to people in the remote segment.

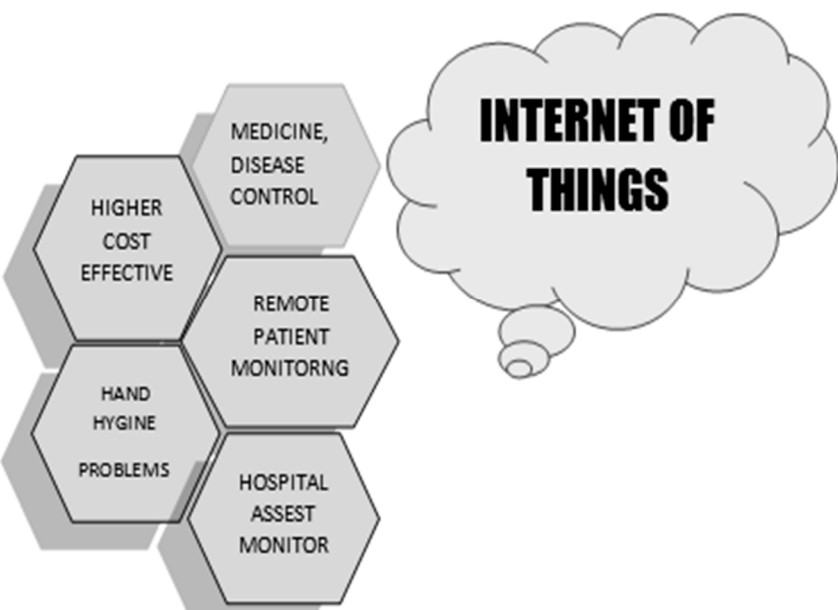

**Figure 6.** IoT Medical services.

Characteristically, IoT is an interconnected system (or network) of physical devices (embedded sensors) linked through a wireless network via the Internet, ensuring the timely exchange of data without human intervention. Before the era of IoT, the interaction of patients with doctors was confined to scheduled appointments and text communications only—there used to be no mechanisms for doctors and hospitals to remotely monitor patients' health and recommend treatments. IoT paved the way for this to happen—more specifically, IoT can be visualised as an inter-play activity between bedside monitors, smart watches, fitness trackers, implanted medical devices and/or hundreds of intelligent electronic devices set up in hospital environment that continuously communicate to each other, performing typical data analytics to ensure risk factor identification, clinical monitoring of patient health conditions, and critical uploading of real time patients' data onto the number of open source health-care cloud platforms empowering patients to manage themselves virtually, i.e., in the physical absence of doctors (Figure 7).

Investigations signifying the deployment of IoT technology in healthcare applications include: IoT-based ultrasound imaging system for diagnosing abnormalities in kidneys [106], IoT-based real time configurable medical imaging equipment [107], utilization of IoT in monitoring, and medical imaging areas of the medical field [108,109]; The IoT healthcare system [110,111] elaborated the role of IoT in early diagnosis and treatment. The KHARE (Kinect HoloLens Assisted Rehabilitation Experience) platform of Microsoft Enterprise Services connecting to Microsoft's Azure IoT Suite is the best example for IoT utilization in healthcare. Smart Fridge by Weka is another application, allowing for the storage of vaccines at optimum temperature levels. A few more outcomes based on explorative IoT research works, directly contributing to the benefits of the healthcare industry, are depicted in Table 5, below.

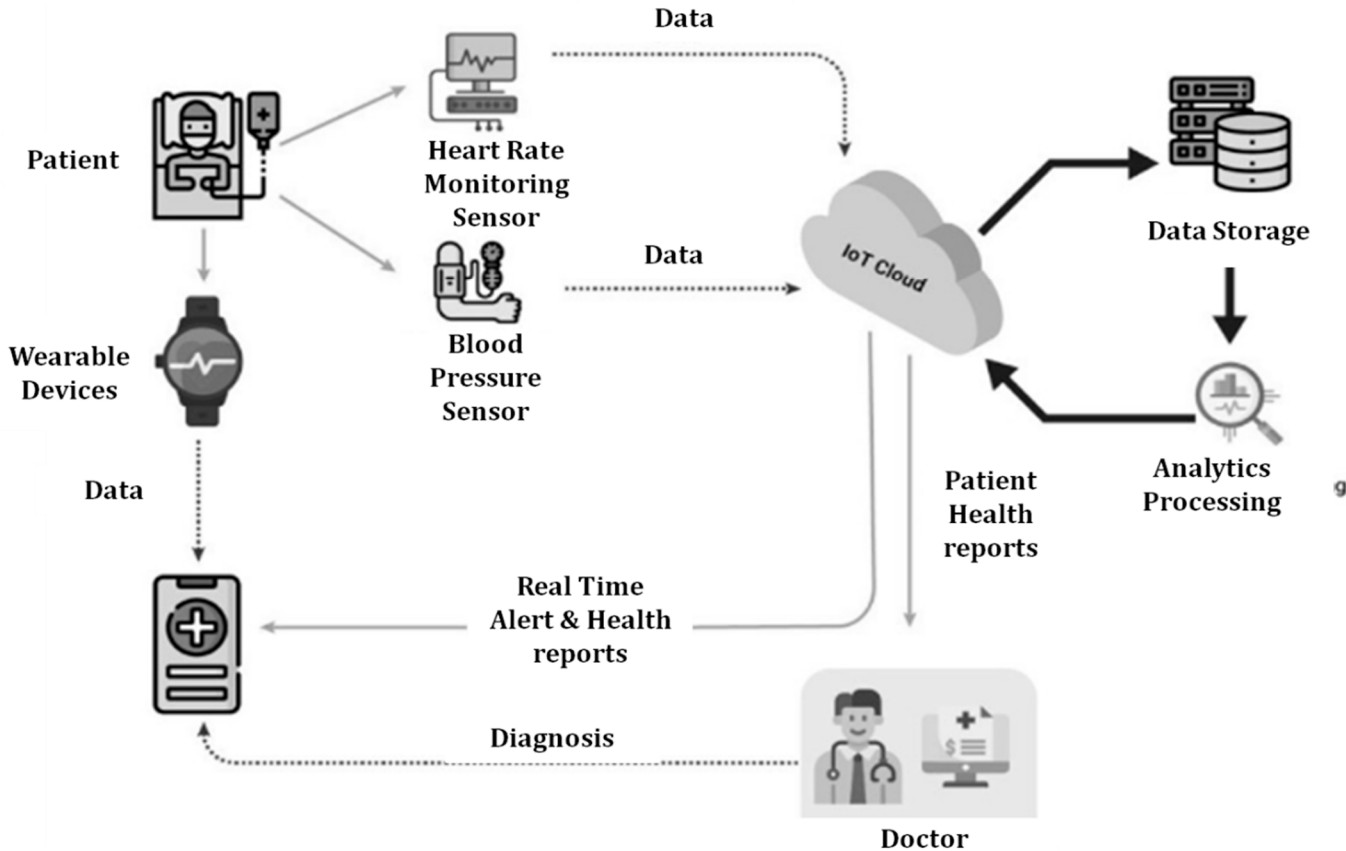

**Figure 7.** Tele-healthcare IoT medical services to remote patients.

**Table 5.** Review of IoT research work contributions to healthcare industry.

| Sl. No | Author | Ref | Category of Health Segment | Application Type | Year |
|--------|--------|-----|-----------------------------|------------------|------|
| 1 | Hussain et al. | [112] | | IoT applications in healthcare devices | 2021 |
| 2 | Iqbal et al. | [113] | | Security framework for IoT-based health applications | 2021 |
| 3 | Bouhassoune et al. | [114] | IoT | Smart patient health monitoring system | 2021 |
| 5 | Paul et al. | [115] | | Remote health monitoring through wearable device and mobile application | 2019 |
| 6 | Halbig et al. | [116] | | Fog computing-based IoT for health monitoring system | 2018 |

Today, Internet of Things (IoT)-enabled devices are empowering physicians and doctors to deliver superlative care through remote monitoring, ensuring enhanced patient satisfaction and drastically reducing the length of patient stay in hospitals, as well as significant cost reductions. The benefits of healthcare IoT technology attributed to the welfare of patients, physicians, hospitals, and insurance companies are highlighted under:

(A)　For patients→ plenty of wireless connected wearable devices such as blood pressure monitoring devices, heart rate monitoring cuffs, glucometers for monitoring insulin levels, oxygen level monitoring devices, calorie counters, steps counters, pace makers, exercise checkers, blood pressure variation monitors, and many types of fitness bands to track the health conditions of aged people, heart patients, and many other bedridden diseased patients—such that even the slightest deviations in the routine course of activities immediately alerts family members and healthcare providers to save their lives.

(B)　For doctors→ With the aid of IoT-based wearable devices, doctors and physicians continuously track patients' health status more effectively and formulate timely treatment plans, ensuring the safety of patient's lives.

(C)　For hospitals→ Almost all IoT-based medical equipment, such as wheelchairs, patient monitors, ventilators, defibrillators, nebulizers, oxygen concentrators, infusion pumps and sterilizers, hygiene monitoring devices, humidity and temperature monitoring equipment, etc., are tagged with sensors to ensure the continuous tracking of real time patient location and data.

(D)　For health insurance companies→ IoT devices created a transparent environment between insurers and customers with regards to pricing, handling claims, underwriting, drafting, and risk assessment procedures to ensure adequate visibility behind all decisions involved. Further health insurance companies are able to eradicate fraud claims based on the data leveraged from intelligent IoT devices and equipment.

According to a global survey, the IoT market for healthcare applications was predicted to exceed around $10 billion by 2024. IoT is now coupled with 5G mobile wireless communications; artificial intelligence and big data technologies significantly revolutionized the healthcare industry by completely transforming the monitoring mechanisms and treatment procedures, as methodologies associated with the patient community globally.

*2.5. VR/AR/MR (HoloLens)*

Virtual Reality (VR) technology provides a simulated reality experience with Complete Medical Immersion into a digital environment to fulfil a range of patient needs; VR technologies are characterized as non-immersive, semi-immersive, or fully immersive VR [117–121]. In fully immersive VR, Head-Mounted Display (HMD) presents a virtual image similar to the real one; in contrast, Augmented reality (AR) provides an immersive experience, in which the real-world content is enhanced by computer-generated three-dimensional content—tied to specific locations, an AR system overlays only the essential information instead of immersing the user totally inside a virtual environment. Significantly, VR/AR technologies support a wide number of healthcare segments (Figure 8)—which include exposure therapy for patients [122], effects of embodiment to deal with body perception disorders due to obesity or anorexia [123], virtual physiotherapy measures enabling patients to recover from surgery [124], and many other simulation tools, such as 360 Proto [125] and Lift-Off [126], are allowing users to create minimal AR/VR prototypes and 3D models.

Google Tiltbrush5 is a unique VR-based virtual modelling tool that enables users to paint and create in VR—based on which users can experiment all kinds of textures and artistic materials to transform any space into canvas. VR Devices such as Oculus Rift (or Google Cardboard) and HTC Vive [117–120] transport users to a variant number of real world and imaginable environments such as VR chats, Volvo test drive reality, etc. On the other hand, Augmented Reality augments the real world by allowing for the integration of digital information, videos and graphics into the users real-world environment; it offers novel approaches in medical education, treatments, and surgery planning etc., clearly elaborating complex medical scenarios to the patients. Further, the expanded utility of the technically interconnected—smart phones, wearables and head mounted display devices—all closely coupled with IoT, Computer vision and AI—greatly revamped the suitability of these technologies in health-care industry. Today, virtual reality is enabling doctors and clinical pathologists to virtually step into a 3600 reconstruction of patient specific inner anatomy and pathology to devise accurately timed surgical plans for each and every patient. Additional investigative contributions depicting the applicability of VR/AR and MR technologies to the healthcare industry are tabulated (Table 6) below for educating the global community of the happening associated with this technology.

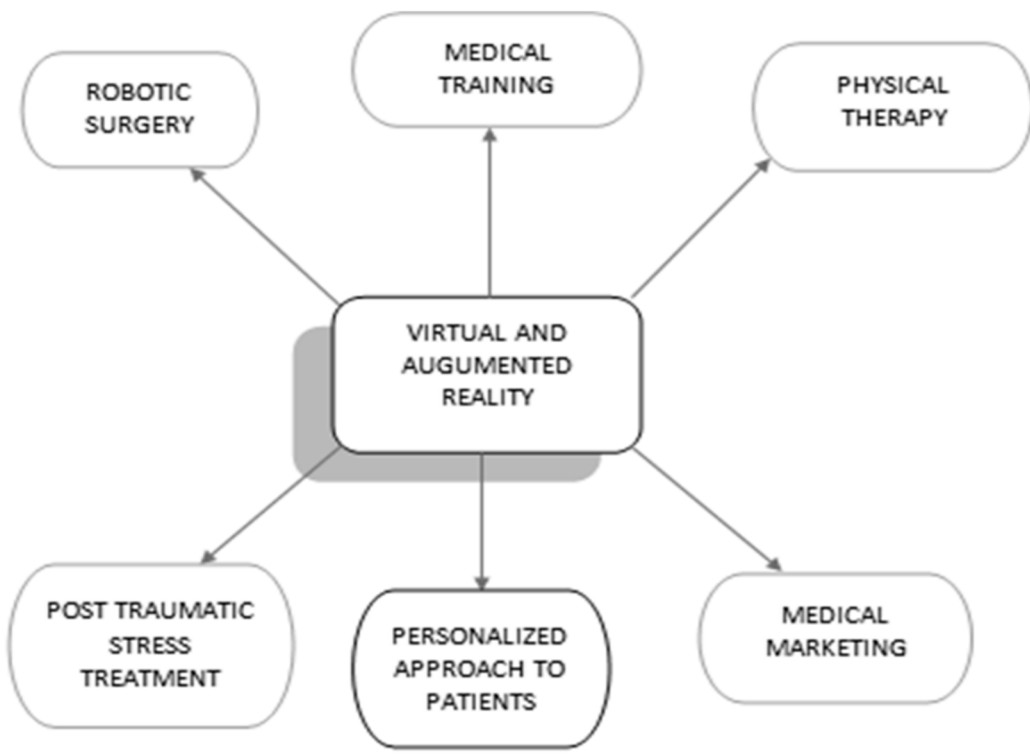

**Figure 8.** Virtual reality deployment in medical field applications.

**Table 6.** Review of VR/AR/MR technologies' deployment in allied medical field applications.

| Sl. No | Author | Ref. | Category of Health Segment | Application Type | Year |
|---|---|---|---|---|---|
| 1 | Coelho et al. | [127] | VR | Challenges of VR in healthcare | 2022 |
| 2 | Morimoto et al. | [128] | AR | AR hybrid simulation model for preoperative planning of surgery | 2022 |
| 3 | Bui et al. | [129] | VR/AR/MR | VR, AR, and MR technology deployment in spine medicine | 2022 |
| 4 | Barsom et al. | [130] | AR | Review of AR technology in tele-mentoring for healthcare. | 2021 |
| 5 | Vidal-Balea et al. | [131] | | An open-source framework of AR technology-based games for paediatric healthcare applications. | 2021 |
| 6 | Jain et al. | [132] | VR | VR hybrid simulation model for endoscopic surgery. | 2020 |
| 7 | Gerup et al. | [133] | AR/MR | Review on AR and MR Technologies for healthcare education. | 2020 |
| 8 | Bouhassoune et al. | [134] | AR | Systematic review of AR technology in medical education. | 2020 |
| 9 | Goo et al. | [135] | VR/MR/AR | 3D Imaging in heart disease: AR, VR, MR and 3D-printing. | 2020 |
| 10 | Amini et al. | [136] | AR/HOLOENS | AR technology-based mastectomy surgical planning using HoloLens template. | 2019 |
| 11 | Condino et al. | [137] | MR/HOLOLENS | Pros and cons of MR-technology-based simulator for orthopaedic surgery using Microsoft Hololens | 2018 |
| 12 | Huang et al. | [138] | AR | Use of AR technology-based glasses in simulation. | 2018 |
| 13 | Lahanas et al. | [139] | | A novel augmented reality simulator for skills assessment in minimal invasive surgery | 2015 |
| 14 | Nomura et al. | [140] | | VR and MR technologies for laparoscopic skill improvement simulator training for medical students. | 2015 |

The further mixed reality technology blends both VR and AR elements, allowing the interaction of real world and digital objects to produce visualizations that unlock new opportunities in the medical field by integrating the [141] mixed reality with medical equipment towards delivering high quality healthcare services to patients. This technology started off with Microsoft HOLOLENS (a notable early MR apparatus) to curtail the risks

associated in performing surgeries and operation times—thereby enabling surgeons to execute risky operations more effectively and safely. Moreover, the advancements in the 5G network together with MR provides prompt clinical services to the people. A few healthcare application areas of VR/AR/MR technologies include:

(A) Surgical training → Medicos, doctors, and surgeons can be trained to perform complex operations with zero surgical errors.

(B) Robotic surgeries → Surgeons equipped with VR technology can perform highly precise operations with the help of robotic devices.

(C) Anxiety and depression treatments → VR Technology is helpful for patients to overcome phobias and stress-induced disorders by resorting to meditations and relaxation treatments, etc.

(D) Hospital navigations → AR-based navigational and way finding tools (Maps) are helpful in locating hospitals, pharmacies, and healthcare centres.

(E) Personalized treatment to patients → VR and MR technologies are helpful to doctors in explaining the surgical procedures and post recovery steps to the patients more elaborately.

(F) AR powered medical education → With AR technology, doctors and surgeons can generate scenario-based 3D representations, ruling out the visualization-based guess works.

(G) Intravenous assistance and vein detection → Doctors and nurses can use handheld augmented reality (AR) devices equipped with laser technology to look "through patient's skin" and deep into their veins to draw the blood.

(H) Defibrillator station way-finding → AR Technology helps patients to detect defibrillator stations in the case of emergency situations.

(I) Smart hospital automation → Smart hospital buildings use AR technology for navigational purposes, linking building models with real time sensors.

(J) Data visualization/body mapping and interactive patient information → AR technology immediately points out the problematic points on the human body with relevant statistical data—with the help of smart glasses connected to the smart-phone.

A few such review articles depicting the application of VR/AR technologies are herewith discussed, i.e., VR simulators for laparoscopic training and surgeries [142–150] and AR surgery looking into real and virtual images displayed on a hologram [151,152]. The latest augmented reality techniques are for surgery. Many other key innovative application areas of surgical simulation include high-resolution imaging diagnostics. Most AR solutions and the emerging mixed reality (MR) technology waive off the conventional surgical procedures [153–159], MR technology-based visceral surgery [160], neurosurgical procedures [161], urinary surgery [162], endoscopy, orthopaedic surgery, and neurosurgery [163,164]. Studies revealed that AR-based surgeries provide enhanced accuracy levels [165–170], and MR-based HoloLens trace out cancerous tissues in patients [171–173]. Holo Anatomy is an advanced AR-powered medical education platform to help professors and doctors in formulating the 3D visualization of human anatomy, including the cardiovascular system, individual organs and their rotary views in different positions and scales, etc. Meanwhile, many use cases (relevant to VR/AR/MR) include pain management, memory retention, medical training, minimally invasive surgeries (laparoscopic surgeries), orthopaedic surgeries, dental surgeries, rehabilitation, and other allied issues relevant to healthcare sector. Continued advancements in these technologies shall definitely enhance the patient outcomes, decrease the length of the hospital stay, decrease the overall expenditures and thereby lead to a prominent decrease in morbidity [112–114,174]. According to the allied market research, it was anticipated that the VR and AR market will reach USD 2.4 Billion by 2026 to revamp and revolutionize the global healthcare sector.

## 3. Future Scope and Challenges

Through robotics, UAV's, 3D-printing, IoT and VR/AR/MR technologies significantly transformed the healthcare sector from conventional patient handling (and treating) mechanisms to today's virtually realistic treatments that are formulated based on the data analytics (sensors' data) and personalised custom-printed 3D devices (patient-centric),

enabling the delivery of superior quality medication and life-saving treatments (through drones) to patients. Several technological limitations adhering to the practical viability in addressing healthcare issues governing various segments of medical technology are left open for the research community to tackle them.

Though artificial intelligence deeply penetrated into the healthcare sector, the essentiality of human surveillance cannot be ignored in the AI-subjected healthcare areas. On the other hand, the applicability of block chain technology in storing patient's electronic health records (EHR) is proven to be economically unviable. Meanwhile, the shared availability of OST (open-source technologies) typically elevates the risk of hacking and data breaches.

(A)     Investigations are deemed essential to measure the effectiveness of robotics' use cases in healthcare services, and pricings relevant to robotic surgeries must be reduced; further training the medical staff involved in using robots is another big challenge. On the other hand, patient's safety, confidentiality, and social attitude towards undergoing treatment with robots are a few addressable challenges.

(B)     The typical usage of drones in the healthcare system suffers from many practical limitations, such as short battery life, low speed of operation, low payload carrying capabilities, improper operations in adverse weather conditions, lack of trained drone pilots, security breaches, social acceptance issues, and technological glitches, etc. The increasing number of no-fly zones in some countries pose limitations on using drones to the line-of-sight distance of the operators—drones are not immune to hijacking, hence their flight is not as secure as it is imagined to be.

(C)     Despite the significance of IoT in medical imaging, the utility of IoT systems in the healthcare sector is bundled with challenges such as complications in large scale dynamic networking, node mobility management, data completeness, data compression, data security, and accurate reproduction of data, etc.; IoT devices are not up to the mark in performing real time monitoring efficiently.

(D)     Though 3D-printing turned out to be a boon for the medical community and healthcare sector, different parts produced by the 3D-printers exhibit variations in geometry and mechanical properties due to the differences in type (and quality) of material, software, calibration, and equipment. On the other hand, the non-eco-friendly process of 3D-printing and materials' scarcity for 3D-printing are few additional challenges to be addressed.

(E)     Though VR/AR/MR technologies channelled a great revolution in the healthcare sector, they carry with them plenty of challenges, such as the heaviness of equipment, inconvenience in usage, discomfort and distraction of wearable devices, and limited battery life, and latency parameters involved in the audio and video signals produce a severe impact in the case of surgeries. On the other hand, there is data privacy, content compatibility, awareness, and portability. The lack of trained operators is a biggest challenge to be addressed.

## 4. Conclusions

The proposed work outlined the significance of disrupting technological advancements such as robotics, drones, 3D-printing, IoT, and VR/AR/MR technologies in addressing many health-care problems, right from general medicine, dentistry, cardiology, neurology, orthopaedics, paediatrics, gynaecology, psychiatry to plastic surgery, etc. Though all these technologies promised to deliver lifesaving treatments, The scientific community ought to address the many issues brought to light as a result of these technologies, although they all carry the possibility of delivering lifesaving medicines.

Though medical robotics traversed a long way to ensure the delivery of safer, efficient medication and treatment to patient communities, in various intercontinental scenarios, the governments shall formulate legislatures governing the general data protection regulations to ensure the effective sharing and utilisation of surgical data among countries, especially while dealing with conflicts of interests arising between governmental and private funding within the countries, as well as between the countries [115,116]. Though drones (UAVs)

symbolize agile tools for the speedy transportation of healthcare products, extensive research works shall be conducted to invent high speed charging batteries, heavy payload carrying drones, high speed drones capable of operating even in adverse weather conditions, etc. Further strong encryption and meticulous product evaluations are needed to prevent the hijacking of drones. Governments should promote installations of more drone centres for immediate replacement and (or) the charging of batteries and also ensure the formulation of drones' utility regulations, as well as encourage aviation authorities to grant legal permissions for drone agencies [20,22,23,25,64–74,100–105,127–140,175–180].

Beyond medical review and validation, medical devices should be approved by the respective country's regulatory agencies, such as, for example, the FDA for the USA, Medicines and Healthcare products Regulatory Agency (MHRA) for the United Kingdom, and the Brazilian Health Regulatory Agency (Anvisa) for Brazil. Hence, it is worth it to say that every single person involved in the production of medical devices must be made aware of his/her legal responsibility, even as a philanthropic action. As far as VR/MR/AR technologies in the healthcare sector are concerned, effective measures to overcome data privacy, content compatibility, and portability includes a wide scale adoption of 5G technology and enabling middleware technology.

**Author Contributions:** Conceptualization, S.S.; methodology, S.S.; software, S.S. and N.K.D.; validation, N.K.D., A.C., S.B. and P.A.V.; formal analysis, N.K.D., A.C., S.B. and P.A.V.; investigation, S.S.; resources, N.K.D. and A.C.; data curation, S.S., N.K.D., A.C. and S.B; writing—original draft preparation, S.S. and N.K.D.; writing—review and editing, N.K.D., A.C., S.B. and P.A.V.; visualization, N.K.D. and A.C.; supervision, N.K.D., S.B. and P.A.V.; project administration, N.K.D., S.B. and P.A.V.; funding acquisition, N.K.D. and A.C. All authors have read and agreed to the published version of the manuscript.

**Funding:** This research received no external funding.

**Institutional Review Board Statement:** Not applicable.

**Informed Consent Statement:** Not applicable.

**Data Availability Statement:** Not applicable.

**Acknowledgments:** The authors would like to acknowledge the timely cooperation and full support extended by the Management of Lendi IET (A) in conducting this review work with the effective utilization of the Central R & D Laboratory facilities at Lendi IET (A).

**Conflicts of Interest:** The authors declare no conflict of interest.

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
