# Peer review of "Technological Advancements and Elucidation Gadgets for Healthcare Applications: An Exhaustive Methodological Review-Part-II (Robotics, Drones, 3D-Printing, Internet of Things, Virtual/Augmented and Mixed Reality)"

_electronics, doi:10.3390/electronics12030548_

Round 1

Reviewer 1 Report

This paper presents the role of technological advancements in the healthcare sector. This paper is divided into two parts: Part I of the review addressed the role played by advanced technologies like artificial intelligence, big data, blockchain, open-source computing, cloud computing, etc. in the healthcare segment. Part II critically reviews the sustainable role of other disrupting technologies like robotics, drones, 3D printing, the Internet of Things, virtual/augmented/mixed reality, etc. in uncovering the vast number of inherent problems encountered by the clinical community.

The presented paper is interesting. However, to publish the work, it needs some changes. My detailed comments are described below.

  1. Please check the text for spelling and technical errors (there are a lot of textual errors in the paper).
  2. How is artificial intelligence changing the medical sciences, and what does the future hold?
  3. How have artificial intelligence technologies permeated the micro-level aspects of health care?
  4. How have augmented and mixed reality technologies enhanced the future of health care?

Author Response

Authors Responses to the Anonymous Reviewers Comments

Reviewer-1

The authors would like to thank the reviewers for the valuable inputs to enhance the quality presentation of the proposed work. The responses to the comments are given below.

Responses are highlighted in Red and Yellow (Grammatical errors and correct references format)

  1. Please check the text for spelling and technical errors (there are a lot of textual errors in the paper).

Answer: Spell check is performed and technical errors addressed are highlighted (Red) in the manuscript.

  1. How is artificial intelligence changing the medical sciences, and what does the future hold?

Answer: The significant contribution of Artificial Intelligence in addressing issues pertaining to medical sciences and relevant investigative explorations governing its applicability in health care domain is discussed in the PART-I Manuscript. While the current PART-II Manuscript is typically highlighting the outcomes resulting from various research contributions governing the applicability of disruptive technologies like Robotics, Drones, 3D Printing, IoT & VR/MR/AR etc., to the health care sector and medical sciences.

However the changing scenario of AI in Medical Science and its future hold was added in the manuscript (Highlighted in RED COLOUR) – The same was indicated below.

“The deployment of Artificial Intelligence (AI) in medical sciences typically augments the tasks of clinicians and healthcare professionals by (say) for example : AI Systems ensure early bird risk mitigation in patients with critical disorders and heart rhythms[18-19], Cancer Predictions[20], nerve and muscle disorders[21], biomedical studies in health care segments[22-27] etc.,  to promote the medical research, enhanced outcome based patient caring at feasible budgets and improved efficiency levels in the life saving hospital administration systems.

The future of AI and supercomputing performances are focused upon accelerated research works uncovering the inhabitancy of pre-diagnostic requirements, surgical hindrances, development of therapeutics by clinical pathologists (surgeons & doctors), innovation of hypersensitive technological gadgets to safeguard people’s health and save lives of patient community”.

  1. How have artificial intelligence technologies permeated the micro-level aspects of health care?

Answer:

Today the applicability of AI Technology in medical sciences and health care is having grass roots covering the micro level aspects of medical sciences and health care sector – which includes clinical diagnosis, data analytics based treatment recommendations akin to AI algorithms outperforming radiologists in spotting malignant tumors.

In medical sciences companies are using AI to understand connection between patients, disease and treatments patient engagement and adherence, AI based VIRTUAL NURSE implemented on Natural Language Processing (NLP) with wireless integration to medical devices and gadgets for providing medical assistance to patients.

(However the Permeability of AI (in health care) to the micro level aspects had been discussed in PART-I MANUSCRIPT) but then -  if at all the anonymous reviewers seek the essentiality of including this in PART-II as well, we shall do the same.

  1. How have augmented and mixed reality technologies enhanced the future of health care?

Today Augmented and Mixed Reality technologies are vastly deployed in medical education, training of healthcare professionals, Pre-surgical, Post-surgical procedures and other health care segments (depicted in Figure: 8 of manuscript) by making the best use of Microsoft HOLOLENS (to curtail the surgical risks & slash down the operation times) enabling surgeons located far apart to execute risky operations more effectively and safely.

The penetration of these technologies in Health care industry shall greatly revolutionize the future health care scenarios to such an extent that today smart phones, wearable’s and head mounted display devices (HOLOLENS) ETC., are closely coupled with IoT, Computer vision & AI technologies etc. Investigations governing the applicability of AR/MR technologies in health care sector are tabulated (in Figure 6 of the manuscript) and Highlighted (RED COLOUR) in the manuscript.The same was indicated below”.

Plenty of review articles depicting application of VR/AR technologies i.e., VR simulators for laparoscopic training and surgeries [101-109], AR surgery looking into real and virtual images displayed on a hologram [110]. Latest augmented reality techniques for surgery [111]. Many other key innovative applications areas of surgical simulation include high-resolution imaging diagnostics. Most AR solutions and emerging mixed reality (MR) technolo­gy to waive of the conventional surgical procedures [112-118], MR technology based visceral surgery [119], neurosur­gical procedures [120], urinary surgery [121], endoscopy, orthopaedic surgery neurosurgery [122-123]. Studies revealed that AR based surgeries provides enhanced accuracy levels [124-128], MR based HoloLens to trace out cancerous tissues in patients [130-132].

Reviewer 2 Report

This study is to present technological advancements and Elucidation Gadgets for healthcare applications. An exhaustive methodological review is provided focused on robotics, drones, 3D-printing, internet of things, virtual/augmented and mixed reality. The topic is timely. Overall study process is  sound and proper. Following are some comments to improve the quality of the manuscript.  

- Title is too long and some words are not directly related to the manuscript. Revise it.  

- In title, 3D-Priniting => 3D-printing

- In the introduction section, a study motivation and study purpose should be clearly addressed. Reflect them accordingly.

- In the conclusion section, add some limitations of the study.

- Enhance the overall manuscript for readers.

Author Response

Authors Responses to the Anonymous Reviewers Comments

Reviewer-2

The authors would like to thank the reviewers for the valuable inputs to enhance the quality presentation of the proposed work. The responses to the comments are given below.

Responses are highlighted in Red and Yellow (Grammatical errors and correct references format)

Following are some comments to improve the quality of the manuscript.  

  1. Title is too long and some words are not directly related to the manuscript. Revise it.  

Answer: The authors would like to inform the anonymous reviewers that – the proposed review work titled   “Technological Advancements and Elucidation Gadgets for Health Care Applications: An Exhaustive Methodological Review-Part-II (Robotics, Drones, 3D-Priniting, Internet of Things, Virtual/Augmented & Mixed Reality)” was undertaken in TWO MANUSCRIPTS (PARTS) i.e., PART-I & PART-II.

We have already submitted PART-I Manuscript (Under Review Now) with same Title. If the reviewers still suggest to go ahead with changing the title – we shall request the PART-I Reviewers as well and change the title for both the manuscripts.

In such a case we propose the below mentioned Title:

Impact of Technological Advancements on Health Care Applications: An Exhaustive Review-Part-II (Robotics, Drones, 3D-Priniting, Internet of Things, Virtual/Augmented & Mixed Reality)” was undertaken in TWO MANUSCRIPTS (PARTS) i.e., PART-I & PART-II

  1. In title, 3D-Priniting => 3D-printing

Answer: Observed and Modified

  1. In the introduction section, a study motivation and study purpose should be clearly addressed. Reflect them accordingly.

Answer: This review work was actually conducted in TWO PARTS (Two Manuscripts Part-I & Part-II), the Part-I Manuscript covered technologies like AI, Bigdata, Block chain Technology, Open source Technologies and Cloud Computing Technologies etc., while technological advancements pertaining to Robotics, Drones, 3D-Printing, IoT (Internet of things), Augmented/Mixed Reality etc. are discussed in the current Part-II Manuscript. Hence the motivation part of the work significantly responsible for conducting this work was discussed in Part-I Manuscript, However identifying the need for including the motivation part responsible for mentioning the said technologies in Part-II Manuscript- we have included the motivation part behind taking up this study in the current Part-II Manuscript as well, (Highlighted RED in manuscript) The same was pasted here for reference

The past few decades witnessed extensive research in the field of healthcare services such that technological advancements are totally indispensable from healthcare domain and its allied segments - apart from the various pulmonary (viral), bacterial and inflammatory diseases (including the deadly COVID-19 pandemic) challenging the medical world critically exposing the limitations of existing health practices, plenty of viruses found in air, water, and soil etc., are causing different infectious diseases from the flu& cold to the deadly COVID-19 pandemic ultimately leading to failure  of multi-organs followed by cardiac arrests [4-7]. The sudden outburst of COVID Pandemic too uncovered the deficiencies of global health-care systems in handling public-health emergency scenarios coupled with plenty of uncovered ailments in clinical microbiology, tuberculosis diagnosis, preoperative surgical planning, Spine Medicine, medical education, mastectomy surgical planning, Orthopaedic Surgery, Laparoscopic Surgery, Surgical Training, Robotic Surgeries, Anxiety and Depression treatments & Hospital Navigations etc.. Motivated the need for identifying the deployment of technological advancements addressing them. This review work is conducted in TWO PARTS highlighting the contribution of technologies like AI, Bigdata, Block-chain Technology, Open Source Technologies and Cloud computing etc in PART-I – while technologies like Robotics, Drones, 3D-Printing, IoT (Internet of things), Augmented/Mixed Reality etc., and their role play in uncovering many health care issues are discussed in the current PART-II manuscript.

  1. In the conclusion section, add some limitations of the study.
    Answer:
    Typically Limitations associated with utility of proposed technologies and their contributions to address many health care issues are discussed in the Future scope and Challenges Section of the manuscript. However the same was modified again and pasted here for the reference of anonymous reviewers.
  2. i) Investigations are deemed essential to measure the effectiveness of robotics use cases in health care services, pricings relevant to robotic surgeries must be slashed down further training the medical staff involved in using robots is another big challenge. On the other hand patient’s safety, confidentiality and social attitude towards undergoing treatment with robots are few addressable challenges.

  1. ii) Typical usage of Drones in health care system suffer from many practical limitations like short battery life, low speed of operation, low payload carrying capabilities, improper operations in adverse weather conditions, lack of trained drone pilots, security breaches, social acceptance issues and technological glitches etc. Increasing number of no-fly zones in some countries pose limitations on using drones to the line of sight distance of the operators - drones are not immune to hijacking hence their flight is not as secure as imagined to be.

  • iii) Despite the significance of IoT in medical imaging – utility of IoT systems in health care sector is bundled with challenges like complications in large scale dynamic networking, node mobility management, data completeness, data compression, data security and accurate reproduction of data etc., - IoT devices are not up to the mark in performing real time monitoring efficiently.

  1. iv) Though 3D Printing turned out to be a boon for medical community and health care sector – different parts produced by 3D printer’s exhibit variations in geometry and mechanical properties due to the differences in type (and quality) of material, software, calibration and equipment. On the other hand the non-eco-friendly process of 3D Printing and materials scarcity for 3D Printing are few additional challenges to be addressed.

  1. v) Though VR/AR/MR technologies channelled great revolution in health care sector– they carry with them plenty of challenges like heaviness of equipment, inconvenience in usage, discomfort and distraction of wearable devices, limited battery life, latency parameters involved in the audio and video signals produce severe impact in case of surgeries. On the other hand data Privacy, content compatibility, awareness, portability. Lack of trained operators is a biggest challenge to be addressed.

  1. Enhance the overall manuscript for readers.

 Answer: In view of reviewer’s suggestions to provide a good reading experience to readers, the author’s have performed various spell check and grammatical corrections and taken utmost care to enhance the quality of the manuscript.

Round 2

Reviewer 1 Report

The authors have satisfactorily addressed the reviewer’s comments. The manuscript needs to be edited for grammar and syntax; please carefully proofread your next submissions, if possible, by a native English speaker.

  1. Please check the text for spelling and technical errors (there are a lot of text errors in the paper).